# Global importance analysis: An interpretability method to quantify importance of genomic features in deep neural networks

Peter K. Koo[1]*, Antonio Majdandzic[1], Matthew Ploenzke[2], Praveen Anand[3], Steffan B. Paul[4]

1 Simons Center for Quantitative Biology, Cold Spring Harbor Laboratory, Cold Spring Harbor, New York, United States of America, 2 Department of Biostatistics, Harvard University, Cambridge, Massachusetts, United States of America, 3 Dana-Farber Cancer Institute, Boston, Massachusetts, United States of America, 4 Bioinformatics and Integrative Genomics Program, Harvard Medical School, Boston, Massachusetts, United States of America

* koo@cshl.edu

**Data Availability Statement:** Dataset and code are available at: http://github.com/p-koo/residualbind.

**Funding:** This work was supported in part by Developmental Funds from the Cancer Center

## Abstract

Deep neural networks have demonstrated improved performance at predicting the sequence specificities of DNA- and RNA-binding proteins compared to previous methods that rely on $k$-mers and position weight matrices. To gain insights into why a DNN makes a given prediction, model interpretability methods, such as attribution methods, can be employed to identify motif-like representations along a given sequence. Because explanations are given on an individual sequence basis and can vary substantially across sequences, deducing generalizable trends across the dataset and quantifying their effect size remains a challenge. Here we introduce global importance analysis (GIA), a model interpretability method that quantifies the population-level effect size that putative patterns have on model predictions. GIA provides an avenue to quantitatively test hypotheses of putative patterns and their interactions with other patterns, as well as map out specific functions the network has learned. As a case study, we demonstrate the utility of GIA on the computational task of predicting RNA-protein interactions from sequence. We first introduce a convolutional network, we call ResidualBind, and benchmark its performance against previous methods on RNAcompete data. Using GIA, we then demonstrate that in addition to sequence motifs, ResidualBind learns a model that considers the number of motifs, their spacing, and sequence context, such as RNA secondary structure and GC-bias.

## Author summary

Although deep neural networks are becoming widely applied in genomics, it remains unclear why they make a given prediction. For model interpretability, attribution methods reveal the independent importance of single nucleotide variants in a given sequence on model predictions. While the resultant attribution maps can help to identify representations of motifs, it remains challenging to identify generalizable patterns across the dataset and to quantify their effect size on model predictions. Here, we introduce an

Support Grant 5P30CA045508 and the Simons Center for Quantitative Biology at Cold Spring Harbor Laboratory (PKK, AM). MP was supported by NIH NCI RFA-CA-19-002. The funders had no role in study design, data collection and analysis, decision to publish, or preparation of the manuscript.

**Competing interests:** The authors have declared that no competing interests exist.

interpretability method called global importance analysis (GIA) to quantify the effect size that putative patterns have on model predictions across a population of sequences. GIA provides a natural follow up to current interpretability methods to quantitatively test hypotheses of putative patterns (and their interactions with other patterns). As a case study, we demonstrate how it can be used for the computational task of predicting RNA-protein interactions and show that deep learning models can learn not only sequence motifs, but also the number of motifs, their spacing, and sequence context, such as RNA secondary structure and GC-bias.

This is a *PLOS Computational Biology* Methods paper.

## Introduction

To infer sequence preferences of RNA-binding proteins (RBPs), a variety of *in vitro* and *in vivo* experimental methods enrich for protein-bound RNA sequences [1–8], and computational methods are used to deduce the consensus RNA sequence and/or structure features that these bound sequences share [9–13]. Many computational approaches employ position-weight-matrices (PWMs) or *k*-mers to model RNA sequence and, in some cases, its secondary structure context. These methods often make simplifying assumptions that do not fully consider biologically important features, such as the multiplicity, size, and position of the features along a given sequence.

Recently, deep neural networks (DNNs), predominantly based on convolutional networks (CNNs) or convolutional-recurrent network hybrids, have emerged as a promising alternative, in most cases, improving prediction performance on held-out test data [13–19]. DNNs are a powerful class of models that can learn a functional mapping between input genomic sequences and experimentally measured labels, requiring minimal feature engineering [20–22]. DeepBind is one of the first "deep learning" approaches to analyze RBP-RNA interactions [13]. At the time, it demonstrated improved performance over PWM- and *k*-mer-based methods on the 2013-RNAcompete dataset, a standard benchmark dataset that consists of 244 *in vitro* affinity selection experiments that span across many RBP families [5]. Since then, other deep learning-based methods have emerged, further improving prediction performance on this dataset [23–25] and other CLIP-seq-based datasets [11, 18, 26, 27].

To validate that DNNs are learning biologically meaningful representations, features important for model predictions are visualized and compared to known motifs, previously identified by PWM- and *k*-mer-based methods [28]. For RBPs, this has been accomplished by visualizing first convolutional layer filters and via attribution methods [13, 18, 23, 24]. First layer filters have been shown to capture motif-like representations, but their efficacy depends highly on choice of model architecture [29], activation function [30], and training procedure [31]. First-order attribution methods, including *in silico* mutagenesis [13, 32] and other gradient-based methods [19, 33–36], are interpretability methods that identify the independent importance of single nucleotide variants in a given sequence toward model predictions—not the effect size of extended patterns such as sequence motifs.

Recent progress has expanded the ability to probe interactions between putative motifs [37–39]. For instance, MaxEnt Interpretation uses Markov Chain Monte Carlo to sample sequences that produce a similar activation profile in the penultimate layer of the DNN [37], allowing for downstream analysis of these sequences. Deep Feature Interaction Maps estimates

the pairwise interactions between features (either nucleotides or subsequences) by monitoring how perturbations of the source features influence the attribution score of the target features in a given sequence [38]. DeepResolve uses gradient ascent to find intermediate feature maps that maximize a class-activated neuron [39]. Class-activated neurons are often highly expressive (i.e. many patterns can drive its high activity) [40], requiring multiple initializations to sample across a diversity of possible patterns that can lead to a similar neuron activity level. The complex optimization landscape makes it difficult to ensure that the feature map space is sampled well enough to capture the diversity of features/interactions learned by a given class-activated neuron.

These aforementioned interpretability methods provide insights into sequence patterns that are associated with model predictions. The feature importances are often noisy and their scores are often meaningful only within the context of an individual sequence, making it challenging to deduce generalizable patterns across the dataset. Nevertheless, these methods provide a powerful approach to derive hypotheses of important patterns such as motifs and putative feature interactions.

Here we introduce global importance analysis (GIA), an approach that enables hypothesis-driven model interpretability to quantitatively measure the effect size that patterns have on model predictions across a population of sequences. GIA is a natural follow-up to current interpretability methods, providing an avenue to move beyond observations of putative features, such as motifs, towards a quantitative understanding of their importance. As a case study, we highlight the capabilities of GIA on the computational task of predicting RNA sequence specificities of RBPs. We introduce ResidualBind, a new convolutional network, and demonstrate that it outperforms previous methods on RNAcompete data. Using GIA, we demonstrate that in addition to sequence motifs, ResidualBind learns a model that considers the number of motifs, their spacing, and sequence context, such as RNA secondary structure and GC-bias.

## Global importance analysis

Global importance analysis measures the population-level effect size that a putative feature, like a motif, has on model predictions. Given a sequence-function relationship *i.e.* $\mathcal{F} : \mathbf{x} \rightarrow \mathbf{y}$, where $\mathbf{x}$ is a sequence of length $L$ ($\mathbf{x} \in \mathcal{A}^L$, where $\mathcal{A} = \{A, C, G, T\}$) and $\mathbf{y}$ represents a corresponding function measurement ($\mathbf{y} \in \mathbb{R}$), the global importance of pattern $\phi$ ($\phi \in \mathcal{A}^l$, where $l < L$) embedded starting at position $i$ in sequences under the *observed* data distribution $\mathcal{D}$ is given by:

$$\mathcal{I}^{\text{global}} \quad = \quad \mathbb{E}_{\mathbf{x}^{\phi_i} \sim \mathcal{D}}[\mathbf{y}|\mathbf{x}] - \mathbb{E}_{\mathbf{x} \sim \mathcal{D}}[\mathbf{y}|\mathbf{x}] \ , \tag{1}$$

where $\mathbb{E}$ is an expectation and $\mathbf{x}^{\phi_i}$ represents sequences drawn from the data distribution that have pattern $\phi$ embedded at positions $[i, i + l]$. Eq 1 quantifies the global importance of pattern $\phi$ across a population of sequences while marginalizing out contributions from other positions. Important to this approach is the randomization of other positions, which is necessary to mitigate the influence of background noise and extraneous confounding signals that may exist in a given sequence. If the dataset is sufficiently large and randomized, then Eq 1 can be calculated directly from the data. However, sequences with the same pattern embedded at the same position and a high diversity at other positions must exist for a good estimate of Eq 1.

Alternatively, a trained DNN can be employed as a surrogate model for experimental measurements by generating data for synthetic sequences necessary to calculate Eq 1, using model predictions as a proxy for experimental measurements. Given a DNN that maps input sequence to output predictions, *i.e. f*: $\mathbf{x} \rightarrow \mathbf{y}^*$, where $\mathbf{y}^*$ represents model predictions, the

estimated global importance of pattern $\phi$ embedded starting at position $i$ under the *approximate* data distribution $\mathcal{D}^*$ is given by:

$$
\begin{aligned}
\hat{\mathcal{I}}^{\text{global}} &= \mathbb{E}_{\mathbf{x}^{\phi_i} \sim \mathcal{D}^*}[\mathbf{y}^*|\mathbf{x}] - \mathbb{E}_{\mathbf{x} \sim \mathcal{D}^*}[\mathbf{y}^*|\mathbf{x}] \ , \\
&\approx \frac{1}{N}\sum_n^N f(\mathbf{x}_n^{\phi_i}) - \frac{1}{N}\sum_n^N f(\mathbf{x}_n) \ ,
\end{aligned}
$$

where $\hat{\mathcal{I}}^{\text{global}}$ represents an estimate of $\mathcal{I}^{\text{global}}$, the expectation is approximated with an average of $N$ samples from an approximate data distribution $\mathcal{D}^* \sim \mathcal{D}$. Without loss of generality, if we sample the same $n$th sequence for both expectations with the only difference being that $\mathbf{x}_n^{\phi_i}$ has an embedded pattern, then we can combine summations, according to:

$$
\hat{\mathcal{I}}^{\text{global}} \approx \frac{1}{N}\sum_n^N \left( f(\mathbf{x}_n^{\phi_i}) - f(\mathbf{x}_n) \right) \ . \tag{2}
$$

The difference between the $n$th sequence with and without the embedded pattern inside the summation of Eq 2 calculates the *local* effect size—the change in prediction caused by the presence of the pattern for the given sequence. The average across $N$ samples estimates the *global* effect size—the change in prediction caused by the presence of the pattern across a population of sequences.

The approximate data distribution must be chosen carefully to be representative of the observed data distribution and to minimize any distributional shift, which can lead to misleading results. Knowing the complete information about the data distribution (including all possible interactions between nucleotides) is intractable, but it is possible to construct a sequence model of the data distribution that preserves some desirable statistical properties. One approach can be to sample sequences from a position-specific probability model of the observed sequences—average nucleotide frequency at each position, also referred to as a profile. A profile model captures position-dependent biases while averaging down position-independent patterns, like motifs. Alternative sequence models include random shuffling and dinucleotide shuffling of the observed sequences, which would maintain the same nucleotide and dinucleotide frequencies, respectively. If there exists high-order dependencies in the observed sequences, such as RNA secondary structure or motif interactions, a distributional shift between the synthetic sequences and the data distribution may arise. Later, we will demonstrate how structured synthetic sequences can be used to address targeted hypotheses of motif dependency on RNA secondary structure. Alternatively, the sequences used in GIA can be sampled directly from the observed dataset, although this requires careful selection such that unaccounted patterns do not persist systematically, which may confound GIA. Prior knowledge can help to select a suitable approximate data distribution. In this paper, we employ GIA using 7 different sampling methods for the approximate data distribution: sampling from a profile model, random shuffle of observed sequences, dinucleotide shuffle of observed sequences, and a random subset of observed sequences sampled from each quartile of experimental binding scores (see Materials and methods).

GIA calculates a statistical association between a sequence pattern and a functional outcome. Similar to randomized control trials, GIA satisfies properties such as ignorability of assignment and exchangeability of treatment effect, *i.e.* which sequences have interventions with embedded patterns, ensuring that GIA provides a causal quantity that is identifiable with Eq 2. Using experimental measurements for the same sequences in our GIA experiments would provide a direct way to calculate causal effect sizes. However, this can be time consuming and costly due to the large number of sequences required to calculate Eq 2 for each

hypothesis. Here, we opt to use a DNN, which has learned to approximate the underlying sequence-function relationship of the data, to "measure" the potential outcome of interventions (*i.e.* embedded patterns)—using predictions in lieu of experimental measurements. Consequently, GIA quantifies the causal effect size of the interventional patterns through the lens of the DNN and is thus subject to the quality of the learned sequence-function relationship. Therefore, GIA is, at its core, a model interpretability tool—a method to quantitatively uncover causal explanations of a DNN.

While Eqs 1 and 2 describe the global importance of a single pattern, GIA supports embedding more than one pattern (as will be demonstrated below). GIA can also be extended to multi-task problems when each class is independent. GIA is a formalization of previous *in silico* experiments that quantify population-level feature importance [28, 32, 41], which helps to distinguish it from other *in silico* experiments to obtain model predictions for query sequences as a proxy for experimental measurements [42] and occlusion-based *in silico* experiments that identify the importance of features local to a sequence under investigation [41, 43].

## Materials and methods

### RNAcompete dataset

**Overview.** We obtained the 2013-RNAcompete dataset from [5], where a full explanation of the data can be found. The 2013-RNAcompete experiments consist of around 241,000 RNA sequences each 38-41 nucleotides in length, split into two sets 'set A' (120,326 sequences) and 'set B' (121,031 sequences). Sequences were designed to ensure that all possible combinations of 9-mers are sampled at least 16 times, with each set getting 8 copies of all possible 9-mers. The provided binding score for each sequence is the log-ratio of the fluorescence intensities of pull-down versus input, which serves as a measure of sequence preference. The 2013-RNAcompete dataset consists of 244 experiments for 207 RBPs using only weakly structured probes [5].

**Preparation of RNAcompete datasets.** Each sequence from 'set A' and 'set B' was converted to a one-hot representation. For a given experiment, we removed sequences with a binding score of NaN. We then performed either clip-transformation or log-transformation. Clip-transformation consists of clipping the extreme binding scores to the 99.9th percentile. Log-transformation processes the binding scores according to the function: $\log(S - S^{MIN} + 1)$, where $S$ is the raw binding score and $S^{MIN}$ is the minimum value across all raw binding scores. This monotonically reduces extreme binding scores while maintaining their rank order, and also yields a distribution that is closer to a Normal distribution. The processed binding scores of either clip-transformation or log-transformation were converted to a $z$-score. We randomly split set A sequences to fractions 0.9 and 0.1 for the training and validation set, respectively. Set B data was held out and used for testing. RNA sequences were converted to a one-hot representation with zero-padding added as needed to ensure all sequences had the same length of 41 nucleotides. Henceforth, all predictions and experimental binding scores are in terms of the $z$-transformed clip- or log-transformed binding score.

### ResidualBind

**Architecture.** ResidualBind takes one-hot encoded RNA sequence as input and outputs a single binding score prediction for an RBP. ResidualBind consists of: (1) convolutional layer (96 filters, filter size 11), (2) dilated residual module, (3) mean-pooling layer (pool size 10), (4) fully-connected hidden layer (256 units), and (5) fully-connected output layer to a single output. The dilated residual module consists of 3 convolutional layers with a dilation rate of 1, 2, and 4, each with a filter size of 3. Each convolutional layer employs batch normalization prior

to a rectified linear unit (ReLU) activation and dropout probabilities according to layers (1) 0.1, (2) 0.2, (4) 0.5. The pre-activated output of the third convolutional layer is added to the inputs of the dilated residual module, a so-called skipped connection [44], the output of which is then activated with a ReLU. The stride of all convolutions is 1 and set to the pool size for the mean-pooling layer. We found that varying the hyperparameter settings largely yielded similar results. Choice of the final model was based on slightly better performance on the validation set.

**Training ResidualBind.** For each RNAcompete experiment, we trained a separate, randomly-initialized ResidualBind model on 'set A' sequences by minimizing the mean squared-error loss function between the model predictions and the experimental binding scores (which were used as labels). All models were trained with mini-batch stochastic gradient descent (mini-batch of 100 sequences) with Adam updates [45] with a decaying learning rate—the initial learning rate was set to 0.001 and decayed by a factor of 0.3 if the model performance on a validation set (as measured by the Pearson correlation) did not improve for 7 epochs. Training was stopped when the model performance on the validation set does not improve for 20 epochs. Optimal parameters were selected by the epoch which yields the highest Pearson correlation on the validation set. The parameters of each model were initialized according to Glorot initialization [46]. On average, it took about 100 epochs (13 seconds/epoch) to train an RNAcompete experiment on a single NVIDIA 2080ti RTX graphical processing unit. Code for building, training, and evaluating ResidualBind was written in Python using Tensorflow 2 [47].

**Evaluation.** Residualbind models were evaluated using the Pearson correlation between model predictions and experimental binding scores on the held-out test data ('Set B'), similar to [12, 13].

**Incorporation of secondary structure profiles.** Paired-unpaired structural profiles were calculated using RNAplfold [48]. Structural profiles consisting of predicted paired probabilities of five types of RNA structure—paired, hairpin-loop, internal loop, multi-loop, and external loop (PHIME)—were calculated using a modified RNAplfold script [10]. For each sequence, the window length (-W parameter) and the maximum spanning base-pair distance (-L parameter) were set to the full length of the sequence. Secondary structure profiles were incorporated into ResidualBind by creating additional input channels. The first convolutional layer now analyzes either 6 channels (4 channels for one-hot primary sequence and 2 channels for PU probabilities) or 9 channels (4 channels for one-hot primary sequence and 5 channels for PHIME probabilities).

## *In silico* mutagenesis

*In silico* mutagenesis is calculated by systematically querying a trained model with new sequences with a different single nucleotide mutation along the sequence and ordering the predictions as a nucleotide-resolution map ($4 \times L$, where 4 is for each nucleotide and $L$ is the length of the sequence). Each prediction is subtracted by the wildtype sequence prediction, effectively giving zeros at positions where the variant matches the wildtype sequence. To visualize the *in silico* mutagenesis maps, a sequence logo is generated for the wildtype sequence, where heights correspond the sensitivity of each position via the L2-norm across variants for each position, and visualized using Logomaker [49].

## Global importance analysis

1,000 synthetic RNA sequences, each 41 nucleotides long, were sampled from 7 different models for the approximate data distributions: 1) randomly sampled from a profile sequence

model; 2) random shuffle of the observed sequences; 3) dinucleotide shuffle of the observed sequences; and 4-7) a random subset of sequences sampled from each quartile of experimental binding scores. Patterns under investigation were embedded in positions specified in each GIA experiment. We queried a trained ResidualBind model with these sequences with and without the embedded pattern. We refer to the difference between the predictions with and without the pattern for each sequence as the "local" importance (the value inside the summation of Eq 2) and the average across the population as the "global" importance.

**Profile sequence model.**   The profile sequence model was generated by averaging the nucleotide frequency statistics across all test sequences. 1,000 synthetic sequences were generated from the profile model by independently sampling the each nucleotide at each position.

**Random shuffle.**   1,000 observed sequences from the test set were randomly chosen and the positions of each sequence was randomly shuffled, thereby preserving the nucleotide frequency while destroying coherent patterns.

**Dinucleotide shuffle.**   1,000 observed sequences from the test set were randomly chosen and the positions of each sequence was dinucleotide shuffled, thereby preserving the dinucleotide frequency while destroying coherent patterns.

**Quartile sampling.**   All observed sequences were sorted according to their experimental binding score and divide into 4 bins. The 1st Quartile corresponds to the sequences with the lowest 25% in binding scores and the 3rd Quartile corresponds to the 50%-75% in binding scores. After this division, we randomly select 1,000 sequences from each bin, creating 4 different sets of sequences from different models of the approximate data distribution.

## Motif visualization

Motif representations learned by ResidualBind are visualized with 2 methods, top $k$-mer motif and $k$-mer alignment motif. Top $k$-mer motif plots the top $k$-mer as a logo with heights scaled according to the L2-norm of the difference in global importance of nucleotide variants at each position, which is measured via GIA by systematically introducing a single nucleotide mutation to the top $k$-mer embedded at positions 18-24, and the global importance of wildtype top $k$-mer.

A $k$-mer alignment-based motif was generated by greedily aligning the top 10 $k$-mers (identified via GIA) to the top $k$-mer according to the maximum cross-correlation value. The nucleotide frequency, weighted by the global importance score for each $k$-mer, gives a matrix that resembles a position probability matrix which can be visualized as a sequence logo using Logomaker [49].

## Results

To demonstrate the utility of GIA, we developed a deep CNN called ResidualBind to address the computational task of predicting RNA-protein interactions. Unlike previous methods designed for this task, ResidualBind employs a residual block consisting of dilated convolutions, which allows it to fit the residual variance not captured by previous layers while considering a larger sequence context [50]. Moreover, the skipped connection in residual blocks foster gradient flow to lower layers, improving training of deeper networks [44]. Dilated convolutions combined with skipped connections have been previously employed in various settings for regulatory genomics [16, 17, 41].

### ResidualBind yields state-of-the-art predictions on the RNAcompete dataset

To compare ResidualBind against previous methods, including MATRIXReduce [9], RNAcontext [10], GraphProt [11], DeepBind [13], RCK [12], DLPRB [23], cDeepbind [24] and

ThermoNet [25], we benchmarked its performance on the 2013-RNAcompete dataset (see Materials and methods for details). We found that ResidualBind (average Pearson correlation: 0.690±0.169) significantly outperforms previously reported methods based on PWMs (MATRIXReduce: 0.353±0.192, RNAcontext: 0.434±0.130), $k$-mers (RCK: 0.460±0.140), and DNNs (DeepBind: 0.409±0.167, cDeepbind: 0.582±0.169, DLPRB: 0.628±0.160, and Thermo-Net: 0.671±0.171, p-value < 0.01, Wilcoxon sign rank test) (Fig 1A). Interestingly, RNAcontext, RCK, ThermoNet, cDeepbind, and DLPRB all take sequence and secondary structure predictions as input, whereas ResidualBind is a pure sequence-based model.

We noticed that the preprocessing step employed by previous methods, which clips large experimental binding scores to their 99.9th percentile value and normalizing to a $z$-score, a technique we refer to as clip-transformation, adversely affects the fidelity of ResidualBind's predictions for higher binding scores, the most biologically relevant regime (Fig 1b). Instead, we prefer preprocessing experimental binding scores with a log-transformation, similar to a Box-Cox transformation, so that its distribution approaches a normal distribution while also maintaining their rank-order (see Materials and methods). With log-transformation, we found that ResidualBind yields higher quality predictions in the high-binding score regime (Fig 1c), although the average performance was essentially the same (Fig 1d, average Pearson correlation is 0.685±0.172 for log-transformation). Henceforth, our downstream interpretability results will be based on preprocessing experimental binding scores with log-transformation.

## Secondary structure context does not help ResidualBind

RNA structure is important for RBP recognition [51]. Previous work, including RCK, RNA-context, DRPLB, cDeepbind, and ThermoNet, have found that including RNA secondary structure predictions as an additional input feature significantly improves the accuracy of their model's predictions. Despite yielding better predictions when considering only sequences, we wanted to test whether incorporating secondary structure predictions would also improve ResidualBind's performance. Similar to previous methods, we predicted two types of RNA secondary structure profiles for each sequence using RNAplfold [48], which provides the probability for each nucleotide to be either paired or unpaired (PU), and a modified RNAplfold script [10], which provides the probability for each nucleotide to be in a structural context: paired, hairpin-loop, internal loop, multi-loop, and external-loop (PHIME). Surprisingly, secondary structure profiles do not increase ResidualBind's performance (Fig 1e and 1f, average Pearson correlation of 0.685±0.172, 0.684±0.183, and 0.682±0.183 for sequence, sequence + PU, and sequence + PHIME, respectively). One possible explanation is that ResidualBind has already learned secondary structure effects from sequence alone, an idea we will explore later.

## Going beyond *in silico* mutagenesis with GIA

It remains unclear why ResidualBind, and many other DNN-based methods, including cDeepbind, DLPRB, and ThermoNet, yield a significant improvement over previous methods based on $k$-mers and PWMs. To gain insights into what DNN-based methods have learned, DLPRB visualizes filter representations while cDeepbind employs *in silico* mutagenesis. Filter representations are sensitive to network design choices [29, 30]; ResidualBind is not designed with the intention of learning interpretable filters. Hence, we opted to employ *in silico* mutagenesis, which systematically probes the effect size that each possible single nucleotide mutation in a given sequence has on model predictions. For validation purposes, we perform a detailed exploration for a ResidualBind model trained on an RNAcompete dataset for RBFOX1 (dataset

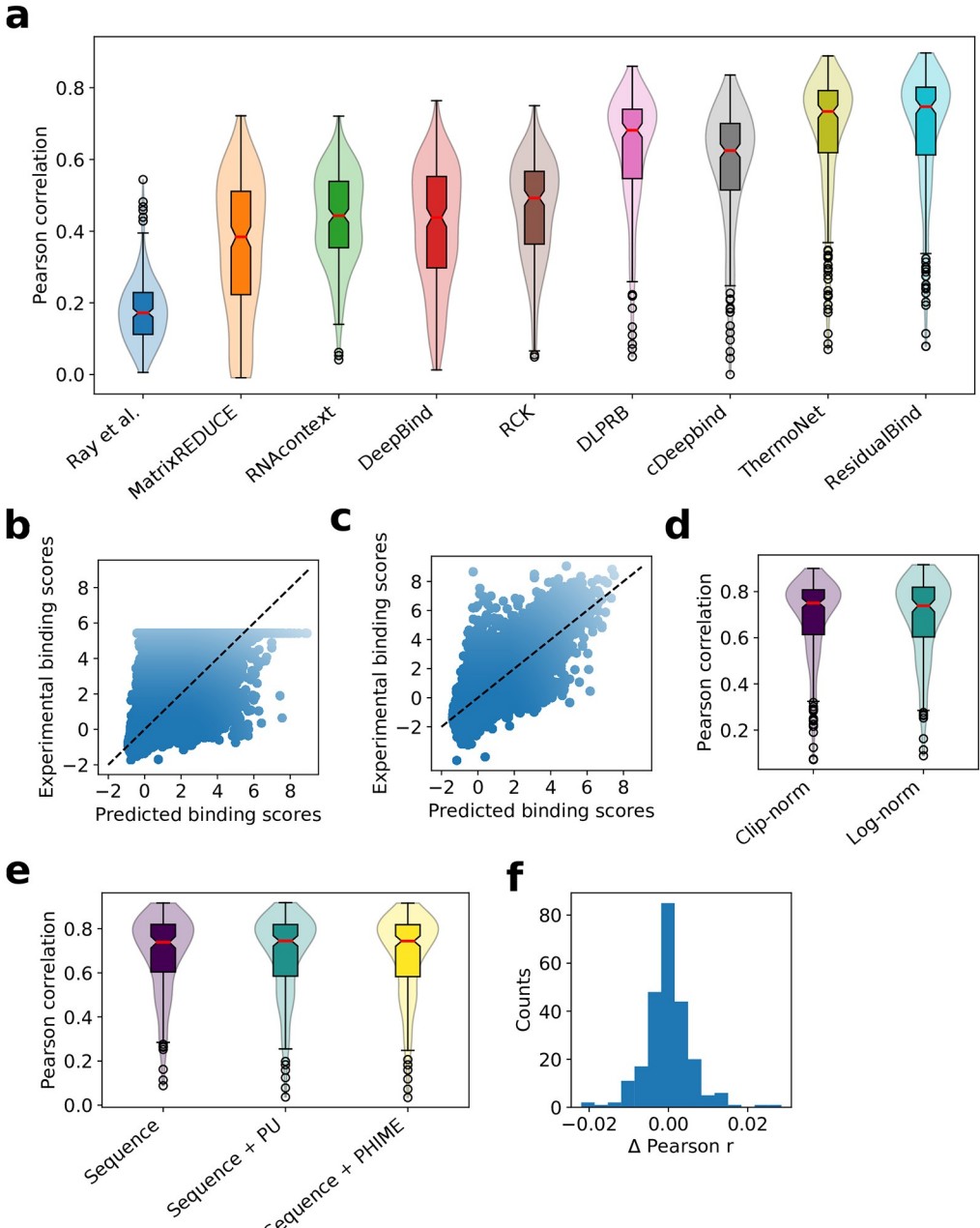

**Fig 1. Performance comparison on the 2013-RNAcompete dataset.** (a) Box-violin plot of test performance by different computational methods. Each plot represents the Pearson correlation between model predictions and experimental binding scores on held out test data for all 244 RBPs of the 2013-RNAcompete dataset. Median value is shown as a red line. (b,c) Scatter plot of ResidualBind's predicted binding scores and experimental binding scores from the test set of an RBP experiment in the 2013-RNAcompete dataset (RNCMPT00169) processed according to (b) clip-transformation and (c) log-transformation. Black dashed line serves as a guide-to-the-eye for a perfect correlation. (d) Box-violin plot of test performance for experimental binding scores processed according to a clip-transformation and a log-transformation. (e) Box-violin plot of the test performance for different input features: sequence, sequence and paired-unpaired secondary structure profiles (sequence+PU), and sequence and PHIME secondary structure profiles (sequence+PHIME). (f) Histogram of the one-to-one performance difference between ResidualBind trained on sequences and trained with additional PHIME secondary structural profiles.

id: RCMPT000168), which has an experimentally verified motif 'UGCAUG' [6, 52, 53]. Fig 2a highlights *in silico* mutagenesis sequence logos for two sequences with high predicted binding scores—one with a perfect match and the other with two mismatches to the canonical RBFOX1 motif (Materials and Methods). Evidently, a single intact RBFOX1 motif is sufficient for a high binding score, while the sequence that contains mismatches to the canonical motif can also have high binding scores by containing several 'sub-optimal' binding sites (Fig 2a, *ii*). This suggests that the number of motifs and possibly their spacing is relevant.

*In silico* mutagenesis, which is the gold standard for model interpretability of DNNs in genomics, is a powerful approach to highlight learned representations that resemble known motifs, albeit locally to an individual sequence. However, it can be challenging to generalize the importance of the patterns that are disentangled from contributions by other factors in a given sequence. Moreover, attribution methods find the independent contribution of each nucleotide on model predictions and hence may not accurately quantify the effect size of larger patterns, such as motifs or combinations of motifs. Therefore, to quantitatively test the hypothesis that ResidualBind learns additive effects from sub-optimal binding sites, we employ GIA.

**GIA shows ResidualBind learns multiple binding sites are additive.** By progressively embedding the canonical RBFOX1 motif (UGCAUG) and a suboptimal motif (AGAAUG, which contains two mismatches at positions 1 and 3) in synthetic sequences sampled from a profile model at various positions, 4-9, 11-16, and 18-23, we find ResidualBind has indeed learned that the contribution of each motif is additive (Fig 2c). We also validate that the spacing between two binding sites can decrease this effect when two motifs are too close (Fig 2d), which manifests biophysically through steric hindrance. While these results are demonstrated for synthetic sequences sampled from a profile model, we found that these results are robust across other models of the approximate data distribution (see S1 Fig).

**GIA identifies expected sequence motifs with *k*-mers.** In many cases, the sequence motif of an RBP is not known *a priori*, which makes the interpretation of *in silico* mutagenesis maps more challenging in practice. One solution is to employ GIA for *ab initio* motif discovery by embedding all possible *k*-mers at positions 18-24. Indeed the top scoring 6-mer that yields the highest importance score for a ResidualBind model trained on RBFOX1 is 'UGCAUG' which is consistent with its canonical motif (Fig 2e). Using the top scoring *k*-mer as a base binding site, we can determine the importance of each nucleotide variant by calculating the global importance for all possible single nucleotide mutations (Fig 2e). Fig 2f shows that the global importance for different variants correlate significantly with experimentally-determined $\ln K_D$ ratios of the variants and wild type measured by surface plasmon resonance experiments [52] (*p*-value = 0.0015, *t*-test). Progressively embedding the top *k*-mer in multiple positions reveals that ResidualBind largely learns a function where non-overlapping motifs are predominantly additive (Fig 2g).

A motif representation can be generated from the global *in silico* mutagenesis analysis in two ways, by plotting the top *k*-mer with heights scaled by the L2-norm of the GIA-based *in silico* mutagenesis scores at each position or by creating an alignment of the top *k*-mers and calculating a weighted average according to their global importance, which provides a position probability matrix that can be converted to a sequence logo. ResidualBind's motif representations and the motifs generated from the original RNAcompete experiment (which are deposited in the CISBP-RNA database [5]) are indeed similar (S1 Table).

**GIA reveals ResidualBind learns RNA secondary structure context from sequence.** The 2013-RNAcompete dataset was specifically designed to be weakly structured [5], which means that the inclusion of secondary structure profiles as input features should, in principle, not add large gains in performance. To better assess whether ResidualBind benefits from the inclusion of secondary structure profiles, we trained ResidualBind on the 2009-RNAcompete

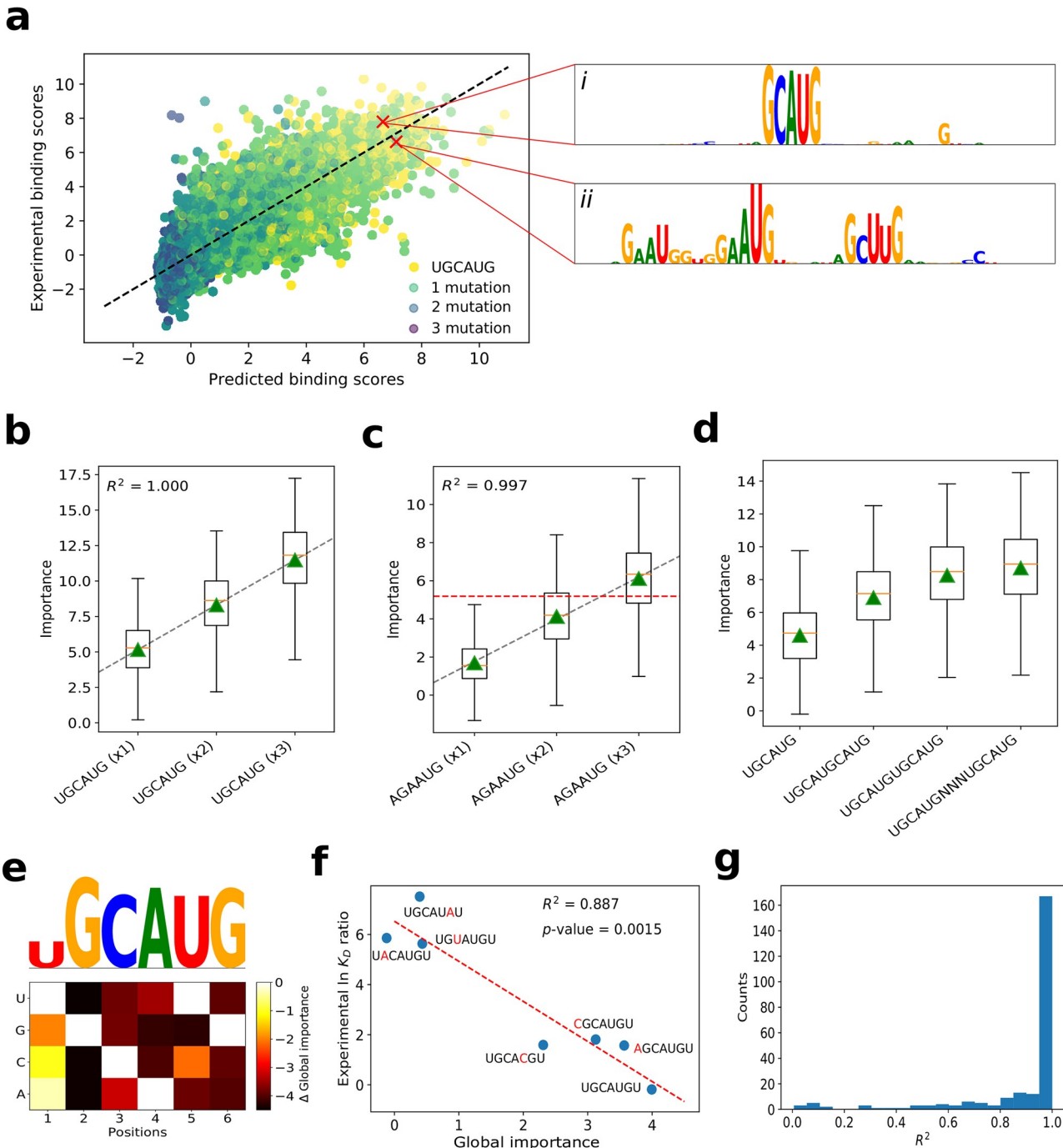

**Fig 2. Investigation of a ResidualBind model trained on RBFOX1.** (a) Scatter plot of experimental binding scores versus predicted binding scores for test sequences in the 2013-RNAcompete dataset for RBFOX1 (Pearson correlation = 0.830). The color of each point is determined by the number of mutations between the canonical motif (UGCAUG) and its best match in the sequence. (i-ii) The inset shows sequence logos for *in silico* mutagenesis maps for a high binding score sequence with at best: (i) a perfect match and (ii) a double nucleotide mismatch to the canonical RBFOX1 motif. Box plot of the local importance for synthetic sequences with varying numbers of the (b) canonical RBFOX1 motif (UGCAUG) and (c) a sub-optimal motif AGAAUG embedded progressively at positions: 4-9, 11-16, and 18-23. Black dashed line represents a linear fit, red horizontal dashed line represents the median, and green triangles represents the global importance. (d) Box plot of the local importance for synthetic sequences with varying degrees of separation between two RBFOX1 motifs ('N' represents a position with random nucleotides). (e) Heatmap of the difference in the global importance for synthetic sequences embedded with single nucleotide mutations of the canonical RBFOX1 motif from wildtype, with a sequence logo that has heights scaled according to the L2-norm at each position. (f) Scatter plot of the experimental ln $K_D$ ratio of the mutant to wild type measured via surface plasmon resonance [51] versus the global importance for the same RBFOX1 variants. Red dashed line represents a linear fit and the $R^2$ and *p*-value from

a *t*-test is shown in the inset. (g) Histogram of the $R^2$ from a linear fit of global importance of embedding different numbers of the top *k*-mer (identified by a separate, *k*-mer-based GIA experiment) at positions: 4-9, 11-16, and 18-23, across the 2013-RNAcompete dataset.

dataset [54], which consists of more structured RNA probes that include stem-loops for nine RBPs. We preprocessed the 2009-RNAcompete dataset in the same way as the 2013-RNAcompete dataset using the log-tranformation for binding scores. On average, ResidualBind yielded only a slight gain in performance by including PU secondary structure profiles (average Pearson correlation of 0.711±0.115 and 0.721±0.116 for sequence only and sequence+PU ResidualBind models, respectively).

In this dataset, VTS1 is a well-studied RBP with a sterile-alpha motif (SAM) domain that has a high affinity towards RNA hairpins that contain 'CNGG' [55, 56]. ResidualBind's performance for VTS1 was comparable (0.6981 and 0.7073 for sequence only and sequence+PU ResidualBind model, respectively), suggesting that the sequence-only model may be learning secondary structure context. An *in silico* mutagenesis analysis for the sequence-only ResidualBind model reveals that the VTS1 motif is found in sequences with a high and low binding score, albeit with flanking nucleotides given significant importance as well (Fig 3a). The presence of a VTS1 motif in a sequence is not sufficient to determine its binding score. Nevertheless, each sequence was accurately predicted by the sequence-only model. The PU secondary structure profile given by RNAplfold for each sequence reveals that the VTS1 motif is inside a loop region of a stem-loop structure in high binding score sequences and in the stem region for low binding score sequences. This further supports that the network may be learning positive and negative contributions of RNA secondary structure context directly from the sequence despite never explicitly being trained to do so. Moreover, the seemingly noisy importance scores that flank the VTS1 motif may represent signatures of secondary structure.

To quantitatively validate that ResidualBind has learned secondary structure context, we performed GIA by embedding the learned VTS1 motif (Fig 3b) in either the loop or stem region of synthetic sequences designed to have a stem-loop structure—enforcing Watson-Crick base pairs at positions 6-16 with 23-33 (Fig 3c). As a control, a similar GIA experiment was performed with the VTS1 motif embedded in the same positions but in random RNAs. Evidently, ResidualBind learns that the VTS1 motif in the context of a hairpin loop leads to higher binding scores compared to when it is placed in other secondary structure contexts. Similarly, these results are robust to choice of model for the approximate data distribution (S2 Fig).

**GIA highlights importance of GC-bias.**   By observing *in silico* mutagenesis plots across many 2013-RNAcompete experiments, we noticed that top scoring sequences exhibited importance scores for known motifs along with GC content towards the 3' end (Fig 4a and 4b). We did not observe any consistent secondary structure preference for the 3' GC-bias using structure predictions given by RNAplfold. Using GIA, we tested the effect size of the GC-bias for sequences with a top 6-mer motif embedded at the center. Fig 4c and 4d show that GC-bias towards the 3' end indeed is a systemic feature for nearly all RNAcompete experiments with an effect size that varies from RBP to RBP (Fig 4e). As expected, consistent results were found across different models of the approximate data distribution (S3–S5 Figs). We do not know the origin of this effect. Many experimental steps in the RNAcompete protocol could lead to this GC-bias [7, 57, 58].

## Discussion

Global importance analysis is a powerful method to quantify the effect size of putative features that are causally linked to model predictions. It provides a framework to quantitatively test

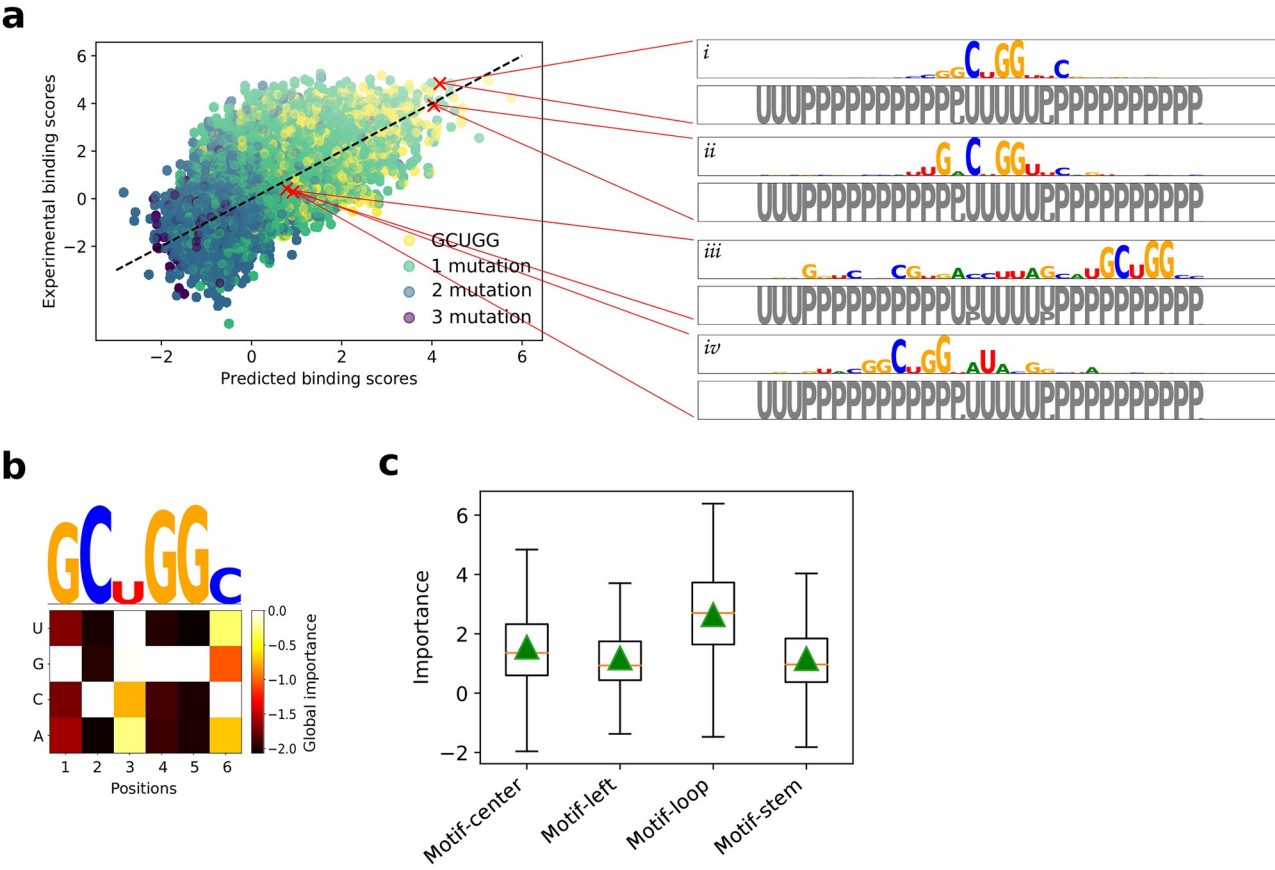

**Fig 3. Investigation of a ResidualBind model trained on VTS1 from the 2009-RNAcompete dataset.** (a) Scatter plot of experimental binding scores versus predicted binding scores on held-out test sequences. The color of each point is determined by the number of mutations between the CISBP-RNA-derived motif (GCUGG) and the best match across the sequence. The inset shows sequence logos for *in silico* mutagenesis maps generated by a ResidualBind model trained only on sequences for representative sequences with high predicted binding scores (i-ii) and low predicted binding scores which contain the VTS1 motif (iii-iv). Below each sequence logo is a PU structure logo, where 'U' represent unpaired (grey) and 'P' represents paired (black), calculated by RNAplfold. (b) Global importance for synthetic sequences embedded with single nucleotide mutations of the top scoring 6-mer (GCUGGC). Above is a sequence logo with heights scaled according to the L2-norm at each position. (c) Box plot of local importance for the top scoring 6-mer embedded in the stem and loop region of synthetic sequences designed with a stem-loop structure and in the same positions in random RNA sequences. Green triangles represent the global importance.

hypotheses of the importance of putative features and explore specific functional relationships using *in silico* experiments, for both positive and negative controls.

As a case study, we introduced ResidualBind for the computational task of predicting RNA-protein interactions. By benchmarking ResidualBind's performance on RNAcompete data, we showed that it outperforms previous methods, including other DNNs. While DNNs as a class of models have largely improved performance compared to previous methods based on PWMs and *k*-mers, model interpretability—based on attribution methods and visualization of first convolutional layer filters—often demonstrate that they learn similar motif representations as previous PWM-based methods, which makes it unclear what factors are driving performance gains. Since first-order attribution methods only inform the effect size of single nucleotide variants on an individual sequence basis, insights have to be gleaned by observing patterns that generalize across multiple sequences. Without ground truth, interpreting plots from attribution methods can be challenging.

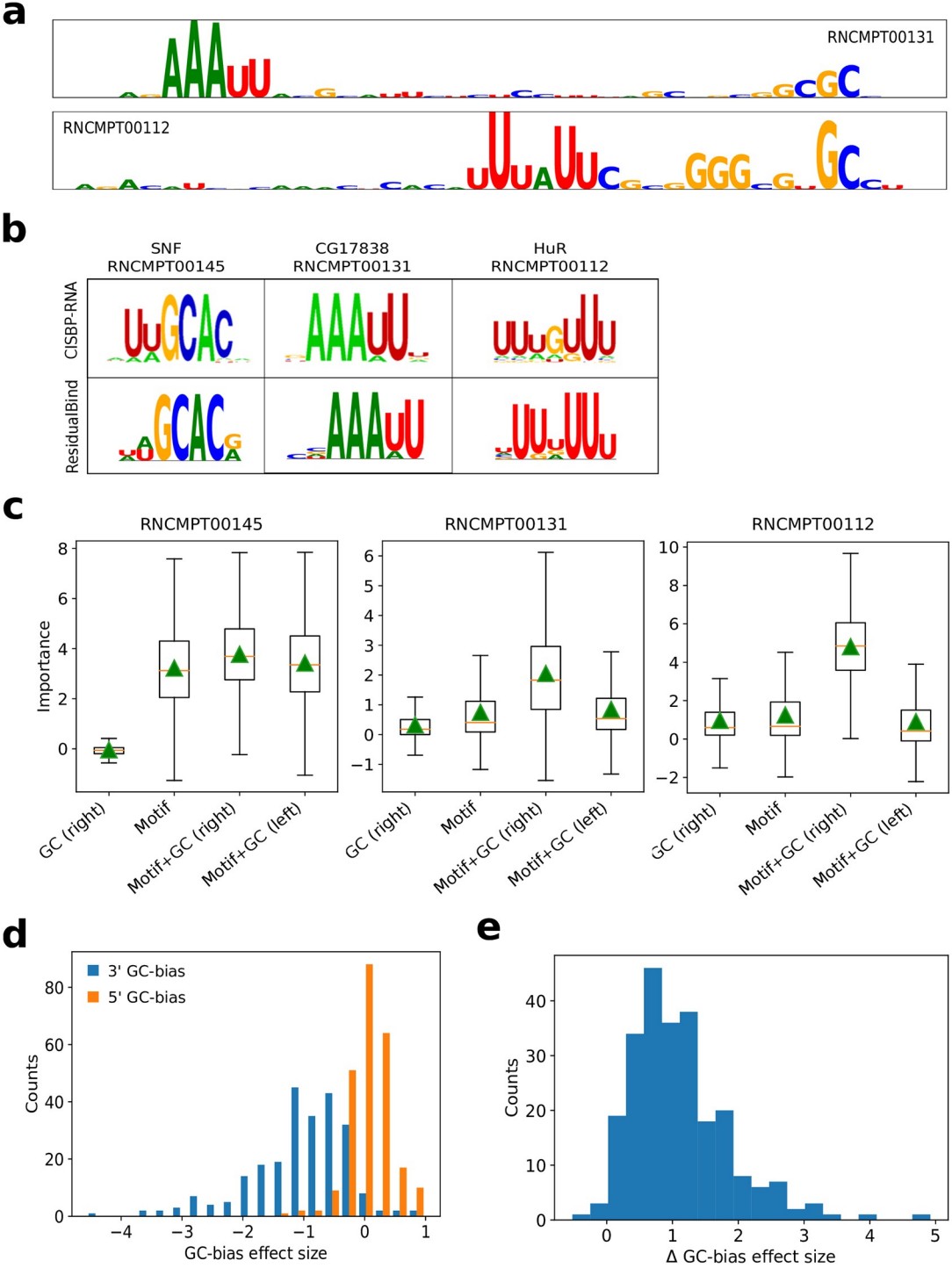

**Fig 4. GC-bias in high binding score sequences.** (a) Representative sequence logos from *in silico* mutagenesis analysis for a test sequence with a top-10 binding score prediction for RNAcompete experiments for CG17838 (RNCMPT00131) and HuR (RNCPT00112). (b) Motif comparison between CISBP-RNA and ResidualBind's motif representations generated by *k*-mer alignments. (c) Box plot of local importance for synthetic sequences with the top scoring 6-mer embedded in position 18-24 and GCGCGC embedded at positions 1-7 (Motif+GC, left) or positions 35-41 (Motif+GC, right). As a control, the GC content embedded at positions 35-41 without any motif is also shown. Green triangles represent the global importance. (d) Histogram of the GC-bias effect size, which is defined as the global importance when GC-bias is placed on the 5' end (orange) and the 3' end (blue) of synthetic sequences with a top scoring 6-mer embedded at positions 18-24 divided by the global importance of the motif at the center without any GC content, for each 2013-RNAcompete experiment. (e) Histogram of the difference between the GC-bias effect size, GC-bias on the 5' end minus the 3' end for each 2013-RNAcompete experiment.

Using GIA, we were able to move beyond speculation from observations of attribution maps by quantitatively testing the relationships between putative features with interventional experiments across a population of sequences. Interestingly, we found that despite Residual-Bind's ability to fit complex non-linear functions, it largely learns an additive model for binding sites, which any linear PWM or $k$-mer based model is fully capable of capturing. We believe the performance gains arise from positional information of the features, including spacing between binding sites and the position of sequence context, such as secondary structures and GC-bias. While these properties are well known features of RBP-RNA interactions, previous computational models were not fully considering these factors, which may have led to their lower performance on the RNAcompete dataset.

## Moving past observational interpretability

Existing model interpretability methods provide a powerful way to identify input features in a given sequence that are important for model predictions. When more than one pattern emerges, it remains challenging to disentangle the relative importance of each feature on model predictions (especially if nonlinear interactions exist) and how sequence context influences this. Associations of putative patterns from observations of attribution maps are useful to generate hypotheses of what the network is learning. GIA provides a downstream analysis that allows one to directly test hypotheses of putative features in a quantitative manner using interventional experiments on synthetic sequences sampled from a model of the approximate data distribution. Of course a hypothesis must be formulated first and so it is still important to perform a thorough first pass analysis with attribution methods and second-order interpretability methods.

## Approximate data distribution

GIA requires choice of sequences in which to embed hypothesis patterns. In the main text, we demonstrate results using a profile model, which is appropriate due to the slight position-specific bias in the 5' end of the RNA probes. There were no significant pairwise frequencies observed for other positions, on average. By design, RNAcompete probes are diverse and thus any random sequence model seems to work well for this dataset. This may explain why GIA was robust across all explored models of the approximate data distribution, including synthetic sequences via shuffling and those that were observed in the data. For other datasets, there may be a more optimal choice. For example, in the binary classification task for ChIP-seq data, the negative label sequences may serve as a suitable model for the approximate data distribution used to embed patterns that will not introduce any significant distributional shift. In practice, it would be prudent to test the robustness of the results across many different models of the approximate data distribution.

## Beyond additive models of interpretability

Previously, quantifying the importance of a motif from attribution maps relied on a strong assumption of an additive model of importance scores [36, 38]. Indeed, attribution methods such as DeepSHAP are explicitly designed to distribute additive contributions of features toward model predictions from a baseline. However, this assumes that each nucleotide's contribution within a motif is additive. GIA can provide the global importance of extended patterns, such as motifs, on model predictions without making such assumptions. Thus, GIA should, in principle, provide more accurate insights when there exist non-additive interactions, i.e. stacking interactions within or flanking motifs and motif-motif interactions [42, 59,

60]. Of course, GIA would yield similar results if the DNN learns an additive model of nucleotide importance within a motif.

## Generalization of GIA

GIA is a general framework that enables one to quantitatively probe the sequence-function relationship learned by a DNN with controlled *in silico* experiments. Such experiments should be done on a case-by-case basis, depending on the hypotheses that one would like to test.

The GIA experiments performed here are specific to the hypotheses generated from analyzing RNAcompete data. Analysis of different datasets will create different hypotheses and thus may require customized GIA experiments. For instance, although the GIA experiments that were performed here embed patterns in specific positions, alternative strategies include embedding the pattern in random positions and marginalizing out this nuisance parameter. This would average over any positional bias. Moreover, GIA does not necessarily require randomizing all input features; it can also be applied to a single sequence context. For instance, occlusion-based experiments can remove a putative feature and be replaced with randomized features. This measures the effect size of the removal of the putative feature(s) while fixing the rest of the sequence context. Unlike previous occlusion-based model interpretability, the importance of the occluded region is marginalized out altogether, thus unaffected by spurious patterns that may arise by chance or non-realistic positions that are effectively "zeroed-out".

Although GIA was developed for genomic sequences, it can be broadly applied to protein sequences and non-sequence data modalities, albeit the approach to randomize input features must be chosen carefully and thus requires domain knowledge.

## ResidualBind

ResidualBind is a flexible model that can be broadly applied to a wide range of different RBPs without modifying hyperparameters for each experiment, although tuning hyperparameters for each experiment would almost certainly boost performance further. While ResidualBind was developed here for RBP-RNA interactions as measured by the RNAcompete dataset, this approach should also generalize to other data modalities that measure sequence-function relationships, including high-throughput assays for protein binding, histone modifications, and chromatin accessibility, given the outputs and loss function are modified appropriately for the task-at-hand.

## *In vitro*-to-*in vivo* generalization gap

Ideally, a computational model trained on an *in vitro* dataset would learn principles that generalize to other datasets, including *in vivo* datasets. However, models trained on one dataset typically perform worse when tested on other datasets derived from different sequencing technologies/protocols [61], which have different technical biases [7, 57, 58, 62]. Learned features like GC-bias may explain why DNNs exhibit large performance gains on held-out RNAcompete data but only a smaller gain compared to *k*-mer-based methods when tasked with generalization to *in vivo* data based on CLIP-seq [23–25]. While we focus our model interpretability efforts on sequences with high binding scores, exploration in other binding score regimes may reveal other sequence context. GIA highlights a path forward to tease out sequencing biases, which can inform downstream analysis to either remove/de-bias unwanted features from the dataset.

## Supporting information

**S1 Table. Performance comparison on RNAcompete.** (Sheet 1) Table shows a comparison of the test performance measured by the Pearson correlation on held out test sequences for different models for each RNAcompete experiment. (Sheet 2) Table shows a comparison of the original RNAcompete motif (represented as a sequence logo) with motif representations learned by ResidualBind (*k*-mer-alignment motif and top *k*-mer motif).
(XLSX)

**S1 Fig. Comparison of different models of the approximate data distribution for multiple binding sites of RBFOX1.** GIA was performed using different models of the approximate data distribution: profile, random shuffle, dinucleotide shuffle, and different binding score quartiles. Box plots of the local importance for synthetic sequences with varying numbers of the canonical RBFOX1 motif (UGCAUG) embedded progressively at positions: 4-9, 11-16, and 18-23. Black dashed line represents a linear fit, red horizontal dashed line represents the median, and green triangles represent the global importance. This demonstrates that GIA is robust across many different models of the approximate data distribution.
(TIF)

**S2 Fig. Comparison of different models of the approximate data distribution for secondary structure preferences of VTS1 from the 2009-RNAcompete dataset.** GIA was performed using different models of the approximate data distribution: profile, random shuffle, dinucleotide shuffle, and different binding score quartiles. Box plot of local importance for the top scoring 6-mer pattern, GCUGGC, embedded in the stem and loop region of synthetic sequences designed with a stem-loop structure and in the same positions in random RNA sequences. Green triangles represent the global importance.
(TIF)

**S3 Fig. Comparison of different models of the approximate data distribution for GC-bias of SNF.** GIA was performed using different models of the approximate data distribution: profile, random shuffle, dinucleotide shuffle, and different binding score quartiles. Box plots show local importance for synthetic sequences with the top scoring 6-mer embedded in position 18-24 and GCGCGC embedded at positions 1-7 (Motif+GC, left) or positions 35-41 (Motif+GC, right). As a control, the GC content embedded at positions 35-41 without any motif is also shown. Green triangles represent the global importance.
(TIF)

**S4 Fig. Comparison of different models of the approximate data distribution for GC-bias of CG17838.** GIA was performed using different models of the approximate data distribution: profile, random shuffle, dinucleotide shuffle, and different binding score quartiles. Box plots show local importance for synthetic sequences with the top scoring 6-mer embedded in position 18-24 and GCGCGC embedded at positions 1-7 (Motif+GC, left) or positions 35-41 (Motif+GC, right). As a control, the GC content embedded at positions 35-41 without any motif is also shown. Green triangles represent the global importance.
(TIF)

**S5 Fig. Comparison of different models of the approximate data distribution for GC-bias of HuR.** GIA was performed using different models of the approximate data distribution: profile, random shuffle, dinucleotide shuffle, and different binding score quartiles. Box plots show local importance for synthetic sequences with the top scoring 6-mer embedded in position 18-24 and GCGCGC embedded at positions 1-7 (Motif+GC, left) or positions 35-41 (Motif+GC, right). As a control, the GC content embedded at positions 35-41 without any motif is also

shown. Green triangles represent the global importance.
(TIF)

## Acknowledgments

The authors would like to thank Sean Eddy, Justin Kinney, David McCandlish, Nicholas Lee, Amaar Tareen, Fred Davis and members of the Koo Lab for either helpful discussions and/or feedback on the manuscript.

## Author Contributions

**Conceptualization:** Peter K. Koo.

**Data curation:** Praveen Anand, Steffan B. Paul.

**Formal analysis:** Peter K. Koo, Praveen Anand, Steffan B. Paul.

**Investigation:** Peter K. Koo, Antonio Majdandzic, Praveen Anand.

**Methodology:** Peter K. Koo, Antonio Majdandzic, Matthew Ploenzke, Steffan B. Paul.

**Software:** Peter K. Koo, Antonio Majdandzic, Praveen Anand.

**Supervision:** Peter K. Koo.

**Validation:** Peter K. Koo, Antonio Majdandzic, Praveen Anand, Steffan B. Paul.

**Visualization:** Peter K. Koo, Antonio Majdandzic, Praveen Anand, Steffan B. Paul.

**Writing – original draft:** Peter K. Koo, Matthew Ploenzke, Praveen Anand.

**Writing – review & editing:** Peter K. Koo, Antonio Majdandzic, Matthew Ploenzke.

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
