## [Decision Letter · Decision Letter 0]

10 Nov 2020

Dear Dr. Koo,

Thank you very much for submitting your manuscript "Global Importance Analysis: A Method to Quantify Importance of Genomic Features in Deep Neural Networks" for consideration at PLOS Computational Biology.

As with all papers reviewed by the journal, your manuscript was reviewed by members of the editorial board and by several independent reviewers. In light of the reviews (below this email), we would like to invite the resubmission of a significantly-revised version that takes into account the reviewers' comments.

We cannot make any decision about publication until we have seen the revised manuscript and your response to the reviewers' comments. Your revised manuscript is also likely to be sent to reviewers for further evaluation.

Sincerely,

Roger Dimitri Kouyos

Associate Editor

PLOS Computational Biology

Weixiong Zhang

Deputy Editor

PLOS Computational Biology

Reviewer's Responses to Questions

**Comments to the Authors:**

Reviewer #1: Koo et al. present a novel algorithm to quantify the effect size, of putative patterns in genomic data, on model predictions (Global Importance Analysis). Further, they developed a new convolutional network to predict RNA-protein interactions (ResidualBind). I would recommend this work for publication with minor revisions.

1. You claim ResidualBind outperforms other methods on predicting RNA-protein interactions. Moreover, you mention predictions of other methods significantly increase when adjusting for secondary structures, while ResidualBind does not benefit. However, your description is not clear, if you compare ResidualBind to other methods while they are adjusted for secondary structures. I.e. e.g. ThermoNet might outperform ResidualBind in an adjusted setting.

If this is the case and ResidualBind already includes the effects of secondary structures, it must miss effects the other methods account for. Could you theorise what these effects could be? If not, please mention (for the ones it applies), that they are already adjusted for secondary structures.

2. In case this provides additional information, could you report the Pearson’s correlation for secondary structures (PU/PHIME), stratified by the probability “high vs low” to form those structures (e.g. median as a cut off or 1st vs 4th quartile)?

3. To further prove the performance boost of ResidualBind and its ability to recognise secondary structures without explicitly teaching them, you could show the performance difference of ResidualBind vs previous methods, using the 2009 set.

However, I leave it to the discretion of the authors.

4. As I understand it, you compare ResidualBind with other methods while preprocessing for all with clip-transformation. While this might be outside of the scope of this work, did you test for the effect of log-transformation on the other methods?

5. Could you clarify if the 2009-RNAcompete Dataset was preprocessed the same way as the 2013-RNAcompete dataset?

6. (Line 305) Add the table number. I assume you mean the supplementary table. Additionally, please provide the source for the CISBP-RNA database.

Reviewer #2: In this paper the authors presented (1) a residual CNN model (ResidualBind) for predicting RNA-protein binding from sequence and (2) a model interpretability approach (Global Importance Analysis) that evaluates the importance of a putative sequence feature by sampling group of sequences from a synthetic distribution and comparing the mean prediction scores of the sequences with and without the putative feature embedded over the population. The authors claimed that this approach enables quantitative hypotheses test and can be used for studying model learned global interactions among patterns and functions such as additive function or sequence context. Although the benchmarking result on RBP predictions for ResidualBind is promising, certain claims regarding the Global Importance Analysis (GIA) are not convincing enough and only supported with ad-hoc synthetic data (which may lack generality) and limited comparison against existing interpretation methods along the same line.

Major concerns:

1)Although the concept of being global and finding summarizing statistics over a population of sequences can be interesting, I’m concerned about the fully synthetic setup and the ad-hoc way the sequences are generated. It seems that the calculation of GIA is highly dependent on the selection of the embedding position i; however, no principled guidance on how to decide i is provided, neither do the rationalization of the choice of location 18-24. Given that the motif can appear in any locations in natural sequences, forcing them to be at a fixed position seems like an un-natural design choice, not to mention the case where there might be position specific patterns and an ad-hoc selected embedding position may easily break that pattern. On the other hand, modeling the contextual distribution is also nontrivial. Although the authors have noticed any uncaptured distribution modes or distribution mismatch could lead to misleading interpretation (especially when there is non-linear dependencies and interaction logic), they have not provided a principled and concrete guidance on how to deal with this problem. Listed options such as using PWM, dinucleotide shuffling can easily fall into the undesired scenario, and again no rationalization is provided why PWM is chosen for the analysis on ResidualBind. I would be more convinced if the authors at least try multiple design choices and empirically analyze their effect on the result.

2)The authors mentioned multiple times that GIA’s global analysis over a population enables the study of feature interactions, however this claim is not well supported by the experiment. The example on counting the number of a repetitive motifs does not involve higher order interactions such as XOR logic or epistatic interaction, and the spacing example is too simple. Although ResidualBind is trained with real experimental data, these synthetic examples (repeating motifs and spacing) may never present in the train data so it is not fair to say that ResidualBind ‘learnt’ to do counting of motifs or spacing. Given that ResidualBind uses mean-pooling instead of max-pooling, it is not surprised to me that simply repeating a motif multiple times will result in higher activation, and one may observe similar correlation by looking at the activation instead of GIA score. More evidence is needed to show that this is something a model learnt and not an artifact of the CNN architecture, and that GIA is necessary for discovery of such pattern.

3)Although the methodology appears to be novel, each of its individual component has overlap with many existing studies, and the literature review and benchmark comparisons on prior methods is insufficient. For example the contrary to sequences without an embedded motif is very similar to integrated gradient [1] and DeepLIFT[2] which compare to a reference sequences. The use of multiple sequence samples to study distributional (instead of individual) feature importance has been introduced in Max-Entropy [3]. Moreover, there exists multiple literatures that explicitly studies feature interactions in genomic neural networks with population level analysis, [4][5][6] and they have involved more comprehensive examples of interactions. The author stated that attribution-based methods cannot provide effect size of extended pattern which is not necessarily true, as the attribution scores of individual nucleotides can be summed up easily to make overall contribution score which has been shown effective in several literatures. It would be more convincing if the author can provide clarifications on distinctions between this work and existing literatures and rationales about why their design choices are superior to these existing methods, as well as quantitative comparison to the existing baselines other than in-silico mutagenesis.

[1] Axiomatic Attribution for Deep Networks, ICML, Sundararajan et. al, 2017

[2] DeepLIFT: Learning Important Features Through Propagating Activation Differences, ICML 2017

[3] Maximum entropy methods for extracting the learned features of deep neural networks, PLOS compbio, 2017

[4] Discovering epistatic feature interactions from neural network models of regulatory DNA sequences, bioinformatics, Greenside et al, 2018

[5] Visualizing complex feature interactions and feature sharing in genomic deep neural networks, BMC Bioinformatics, Liu et al, 2019

[6] Deep learning at base-resolution reveals motif syntax of the cis-regulatory code, bioarxiv, Avsec et. al, 2020

4)It seems that the application of GIA requires prior knowledge about the location and the sequence which needs to be analyzed. Although it is helpful for validating putative features, it may not be very useful in general cases where the underlining mechanism and feature syntax are unknown. The author gave one example on an initio motif discovery; however, this is only applicable to single motif with known length and not easily generalizable to other complex features and interactions.

Minor concerns

1)The author referred to a bioarxiv literature which appears to be largely overlapping with the current submission and should be considered as the prior version of same submission. It would be better to remove such reference as it is confusing.

2)For the motif visualization of top GIA k-mer with in-silico mutagenesis, it is unclear if the L2-norm is taken w.r.t the GIA score or the change of GIA score after introducing mutation. It would make more sense if it’s the latter one, as an L2 norm of GIA score of all variants does not have information specific to the referenced wildtype and thus do not contain information about that nucleotide’s sensitivity. Also the scale of global importance values on fig 2e (-4,0) is difference than that of fig 2f (0,4).

3)There are several typos/latex compilation errors in the text, e.g. line 222 p-value?0.01, and line305 missing table.

**Have all data underlying the figures and results presented in the manuscript been provided?**

Reviewer #1: Yes

Reviewer #2: Yes

PLOS authors have the option to publish the peer review history of their article (what does this mean?). If published, this will include your full peer review and any attached files.

Reviewer #1: No

Reviewer #2: No
---

## [Decision Letter · Decision Letter 1]

10 Mar 2021

Dear Dr. Koo,

Thank you very much for submitting your manuscript "Global Importance Analysis: An Interpretability Method to Quantify Importance of Genomic Features in Deep Neural Networks" for consideration at PLOS Computational Biology. As with all papers reviewed by the journal, your manuscript was reviewed by members of the editorial board and by several independent reviewers. The reviewers appreciated the attention to an important topic. Based on the reviews, we are likely to accept this manuscript for publication, providing that you modify the manuscript according to the review recommendations.

Sincerely,

Roger Dimitri Kouyos

Associate Editor

PLOS Computational Biology

Weixiong Zhang

Deputy Editor

PLOS Computational Biology

[LINK]

Reviewer's Responses to Questions

**Comments to the Authors:**

Reviewer #1: The authors addressed all my concerns. I recommend the manuscript for publication

Reviewer #2: I appreciate that the authors have improved the manuscript based on prior reviews and most of the concerns are addressed.

Specifically, the authors added clarifications in the discussion that GIA mainly serves as tool for downstream analysis of other interpretability methods. This is important as it helps the audience to identify which part of the practice in this paper is “artificially chosen” and needs customization in practice. Thus, it might be better if the authors could make this distinction clear even earlier in the paper (perhaps in introduction/method) whenever ‘ad-hoc’ design choices are used. I

t is good that the authors included 6 additional strategies for sampling synthetic sequences and showed more extensive results using max-pooling. I also appreciate the more comprehensive literature review on pre-existing methods of attribution-based and interaction interpretability tools, although the description of [39] is a bit misleading. As far as I know, DeepResolve uses multiple optimization samples and not just “one optimization run” to explore diverse patterns in the optimization landscape, and thus the limitation the author mentioned could be largely alleviated. It would be better if the authors consider modification of the corresponding paragraph for correctness.

**Have all data underlying the figures and results presented in the manuscript been provided?**

Reviewer #1: Yes

Reviewer #2: Yes

PLOS authors have the option to publish the peer review history of their article (what does this mean?). If published, this will include your full peer review and any attached files.

Reviewer #1: No

Reviewer #2: No

Figure Files:

Data Requirements:

Reproducibility:

References:

---

## [Editor Report · Decision Letter 2]

30 Mar 2021

Dear Dr. Koo,

We are pleased to inform you that your manuscript 'Global Importance Analysis: An Interpretability Method to Quantify Importance of Genomic Features in Deep Neural Networks' has been provisionally accepted for publication in PLOS Computational Biology.

Best regards,

Roger Dimitri Kouyos

Associate Editor

PLOS Computational Biology

Weixiong Zhang

Deputy Editor

PLOS Computational Biology

---

## [Editor Report · Acceptance letter]

20 Apr 2021

PCOMPBIOL-D-20-01489R2 

Global Importance Analysis: An Interpretability Method to Quantify Importance of Genomic Features in Deep Neural Networks

Dear Dr Koo,

I am pleased to inform you that your manuscript has been formally accepted for publication in PLOS Computational Biology. Your manuscript is now with our production department and you will be notified of the publication date in due course.

With kind regards,

Andrea Szabo
